# Capability, Opportunity, and Motivation—Identifying Constructs for Increasing Physical Activity Behaviours in Women with Polycystic Ovary Syndrome (PCOS)

**DOI:** 10.3390/ijerph20032309

**Published:** 2023-01-28

**Authors:** Chris Kite, Lou Atkinson, Gordon McGregor, Cain C. T. Clark, Harpal S. Randeva, Ioannis Kyrou

**Affiliations:** 1School of Public Health Studies, Faculty of Education, Health and Wellbeing, University of Wolverhampton, Wolverhampton WV1 1LY, UK; 2Warwickshire Institute for the Study of Diabetes, Endocrinology and Metabolism (WISDEM), University Hospitals Coventry and Warwickshire NHS Trust, Coventry CV2 2DX, UK; 3Centre for Sport, Exercise and Life Sciences, Research Institute for Health & Wellbeing, Coventry University, Coventry CV1 5FB, UK; 4Warwick Medical School, University of Warwick, Coventry CV4 7AL, UK; 5EXI, People’s Mission Hall, 20-30 Whitechapel Road, London E1 1EW, UK; 6Department of Cardiopulmonary Rehabilitation, Centre for Exercise & Health, University Hospitals Coventry & Warwickshire NHS Trust, Coventry CV2 2DX, UK; 7Centre for Intelligent Healthcare, Coventry University, Coventry CV1 5FB, UK; 8Aston Medical School, College of Health and Life Sciences, Aston University, Birmingham B4 7ET, UK; 9Laboratory of Dietetics & Quality of Life, Department of Food Science & Human Nutrition, School of Food & Nutritional Sciences, Agricultural University of Athens, 11855 Athens, Greece

**Keywords:** exercise, quality of life, behaviour change, health promotion, intervention design

## Abstract

Polycystic ovary syndrome (PCOS) is the commonest endocrinopathy in reproductive-aged women. Because increased adiposity is pivotal in the severity of PCOS-related symptoms, treatment usually incorporates increasing energy expenditure through physical activity (PA). This study aimed to understand the reasons why women with PCOS engage in PA/exercise, which could support the development of targeted behavioural interventions in this at-risk population. Validated questionnaires were administered for self-reported PA levels, quality of life, mental health, illness perception, sleep quality, and capability, opportunity, and motivation (COM) for PA. Using categorical PA data, outcomes were compared between groups; ordinal logistic regression (OLR) was used to identify whether COM could explain PA categorisation. A total of 333 participants were eligible; favourable differences were reported for body mass index, depression, mental wellbeing, self-rated health, illness perception, and insomnia severity for those reporting the highest PA levels. COM scores increased according to PA categorisation, whilst OLR identified conscious and automatic motivation as explaining the largest PA variance. The most active participants reported favourable data for most outcomes. However, determining whether health is protected by higher PA or ill health is a barrier to PA was not possible. These findings suggest that future behavioural interventions should be targeted at increasing patient motivation.

## 1. Introduction

Affecting up to 10–15% of reproductive-aged women, polycystic ovary syndrome (PCOS) represents the most common endocrinopathy in this population [1,2]. Following exclusion of other diseases with similar presentation, a PCOS diagnosis is made based upon the presence of at least two out of three principal characteristics, namely (i) menstrual dysregulation (chronic oligo-/anovulation), (ii) hyperandrogenism that is either clinical (i.e., hirsutism, acne, or androgenic alopecia) or/and biochemical (i.e., elevated testosterone/androgens), and (iii) polycystic ovaries on ultrasound scanning [3,4]. Notably, women with PCOS further exhibit high prevalence of overweightness/obesity, with up to 88% of women with PCOS exhibiting increased adiposity (particularly centrally), and a body mass index (BMI) above the normal range [5], which is often associated with further cardiometabolic comorbidities, such as insulin resistance, type 2 diabetes (T2DM), hypertension, dyslipidaemia, and obstructive sleep apnoea (OSA) [6,7,8]. In addition, PCOS is also associated with psychological comorbidity, increasing the risk of anxiety, depression, and chronic stress [9,10], with the resultant health burden having a detrimental effect on overall quality of life (QoL) [11,12].

Although defining the exact aetiology of PCOS has proved elusive [13], the aforementioned increased adiposity is considered a pivotal factor in the presentation and severity of PCOS-related symptoms [14]. Thus, current treatment recommendations for PCOS focus on lifestyle modification, aiming at weight loss [15]. Indeed, similarly to what is noted for T2DM [16], even moderate weight loss (e.g., 5–10%) in women with PCOS has been shown to have clinically important benefits to insulin resistance, menstrual regularity, the severity of hyperandrogenaemia and QoL [17,18,19]. Accordingly, first-line PCOS treatment recommendations involve lifestyle changes, as well as behaviour change strategies, focusing on dietary control and increased energy expenditure through exercise and/or physical activity (PA) interventions [20,21]. Regarding the latter, a systematic review and meta-analysis by our group [22] showed a statistically beneficial change from baseline for insulin sensitivity, lipid profile, waist circumference, and cardiorespiratory fitness (VO_2_ max/peak) in women with PCOS completing exercise interventions, when compared to those in a control group. Similarly, a scoping review revealed potentially beneficial effects of resistance training programmes on a range of pertinent outcomes (e.g., glycaemic control, body composition, and androgens) in women with PCOS [23]. However, the findings of our systematic reviews also highlighted that research regarding exercise/PA interventions in women with PCOS remains limited and that well-designed studies are still needed in this field. In this context, gaps in the existing relevant research remain regarding the most appropriate behaviour change strategies to support women with PCOS to increase their PA levels. The lack of such PCOS-specific data became even more apparent during the COVID-19 pandemic, when enforced lockdown measures restricted the exercise opportunities for the general population, including women with PCOS, who in addition are potentially at increased risk for more severe COVID-19 [24,25]. Indeed, women with PCOS may exhibit higher risk for more severe COVID-19 due to frequent comorbidities, such as obesity, type 2 diabetes and hypertension, whilst hyperandrogenism and other potential features of PCOS (e.g., immune dysfunction and a low-grade chronic inflammatory state) are also hypothesized to increase the risk of severe COVID-19 [24,26,27,28].

Given the experiences regarding lockdown and the associated movement restrictions during the COVID-19 pandemic, better understanding of the reasons regarding why women with PCOS do, or do not, participate in PA/exercise is expected to allow the development of targeted behavioural interventions specifically designed to promote PA in this at-risk population. A common theoretical framework to aid the understanding of health-related behaviour change is the Capabilities, Opportunities, Motivations, and Behaviour (COM-B) model [29]. The COM-B model incorporates three essential conditions that, when measured, can explain the variance in a given behaviour. In the context of the current study: (1) Capability concerns an individual’s physical and psychological capacity to engage in PA; (2) Opportunity relates to external factors (social or physical) that either prompt PA or allow the individual to engage in PA; (3) Motivation, separated into automatic and reflective, is defined as cognitive processes that energise and direct PA behaviours [29]. This model also purports that one’s capability or opportunity can influence motivation, and, whilst collectively they may influence PA, a bi-directional relationship exists where engaging in PA can also alter an individual’s capability, opportunity, and motivation [30]. Accordingly, it was hypothesised that women who self-reported being more physically active perceived their health and QoL to be better than those who were less active. Once the null-hypothesis was rejected, the objective was to ascertain whether constructs of the COM-B model [29] can be used to predict variance in PA behaviour so that these individual constructs could be targeted in future behavioural interventions to increase PA in women with PCOS.

## 2. Materials and Methods

A cross-sectional study, which used a web-based set of structured and validated questionnaires, was conducted between 2 June and 17 August 2020, following ethical approval from Coventry University (P106195). Given the national lockdown for COVID-19, a convenience sample of participants was recruited via advertisements distributed using social media, PCOS support groups on Facebook, and with support from Verity (the UK PCOS charity). Eligible participants had to be adult females (age: 18–45 years) residing in the UK who had a prior diagnosis of PCOS. By contrast, exclusion criteria were women without a PCOS diagnosis, those outside the specified reproductive age (i.e., <18 years or >45 years), and those not residing in the UK.

Participants could request a URL link to the survey directly from a member of the research team, or they could access via the study adverts online, which contained a clickable URL link. The URL directed the participants to the participant information sheet and allowed them to confirm informed consent and eligibility confirmation before transferring them to the study questionnaires. All questionnaires were completed online using the survey software Qualtrics© XM (Qualtrics XM, Provo, UT, USA). Failure of participants to complete the entire survey was construed as withdrawal from the study, and accordingly we also excluded those participants. In total, 614 participants confirmed their eligibility to participate in the study; however, 281 participants did not fully complete the online survey, thus withdrawing their consent to participate, and so were excluded from the study.

A demographics questionnaire was completed by all participants to capture relevant participant characteristics; selected pertinent data are reported in Table 1, whilst the remainder have previously been reported [25]. To assess PCOS-specific QoL, the PCOSQOL was utilised [31], which is a previously validated QoL measure for UK-based women with PCOS and is the first PCOS-specific tool that accounts for all phenotypes of PCOS according to the most recent diagnostic criteria [31]. This questionnaire uses 35 Likert-based questions and asks participants to rate the impact of their PCOS upon their day-to-day life. The individual scores are summed to provide a total measure of QoL (higher scores indicate better QoL), but there are also four subscales that assess the *Impact of PCOS*, *Infertility*, *Hirsutism*, and *Mood* on QoL [32,33,34].

Information about certain mental health aspects of the participants was collected using the Depression, Anxiety, and Stress Scale (DASS-21) [35]. The DASS-21 questionnaire asks participants to respond to seven questions for each domain (i.e., depression, anxiety, and stress) by rating their agreement (0–3) to a series of statements, whilst domain scores are summed with a higher score indicating a greater severity of morbidity. In addition, the 14-item Warwick–Edinburgh Mental Well-being Scale (WEMWBS) was used to measure subjective wellbeing and psychological functioning [36]. The WEMWBS is scored by summing participant responses to each Likert scale-based question; scores can range from 14-70, with a higher score indicative of higher wellbeing and psychological functioning.

Participants were also asked to rate their ‘*health today*’ using the EQ visual analogue scale (VAS) from the 5-level EQ-5D version [37]. The EQ VAS asks the participants to score their health on a vertical VAS that labels the endpoints as ‘*the best health you can imagine*’ (100) and ‘*the worst health you can imagine*’ (0), thus forming a scaled score [38]. Furthermore, to understand more about how women with PCOS perceived their illness, participants were asked to complete the Brief Illness Perception Questionnaire (IPQ-B) [39]. The IPQ-B asks participants to score (from 0–10) each dimension: *perceived consequences*, *timeline* (acute-chronic), *amount of perceived personal control*, *treatment control*, *identity* (symptoms), *concern about the illness*, *coherence of the illness,* and *emotional representation*; the individual dimension scores are then summated, with a higher score indicating stronger (negative) perceptions [39].

Moreover, to understand self-perceived insomnia, the Insomnia Severity Index (ISI) was used [40], which has been shown to be a valid and reliable tool for measuring sleep quality [41,42]. The ISI consists of seven items that evaluate the severity of sleep disruption, and the degree to which day-to-day life is affected. Participants are asked to consider the previous two weeks and then to rate each of the seven items on a five-point Likert scale. Total scores range from 0–28, with higher scores representing greater insomnia severity.

Finally, to capture self-reported participant PA levels, the 9-item International Physical Activity Questionnaire (IPAQ) short form was used. Using a last seven-day recall, the IPAQ captures PA at four intensity levels based upon metabolic equivalents of task (METs) [43]: vigorous-intensity (≥6 METs), moderate-intensity (3–5.9 METs), walking (3.3 METs), and sitting (1 MET). The IPAQ allows researchers to calculate weekly MET-minutes and daily sitting time as continuous scores, but also to categorise participants (low, moderate, high levels of PA) based on their reported activity levels. Considering PA as a health protective behaviour, we utilised a previously validated generic measure of capabilities, opportunities, and motivations [44] to quantify the extent to which the COM-B domains [29] can predict PA levels. The generic questionnaire itself consists of six Likert-type questions (0 to 10) that task respondents to rate their agreement to a series of statements that assess perceptions of each of the six subdomains. The questionnaire enables researchers to replace generic text with bespoke text relating to the health behaviour being investigated; thus, for the purposes of this study, we substituted ‘*change my behaviour’* for ‘*to do enough physical activity’* (Appendix A).

### Statistical Analysis

Statistical analysis was completed in jamovi (version 1.8), and statistical significance was set at *p* < 0.05. Shapiro–Wilk tests of normality were completed on outcome data, both for the whole dataset and individually based upon categorical IPAQ responses. Due to the prevalence of non-normally distributed data, a non-parametric approach was used for all statistical analyses.

IPAQ PA classification of participants was used as a categorical variable in analyses, and Kruskal–Wallis one-way analysis of variance was completed to determine whether there were statistical differences between the three study groups (Table 2). In variables where statistical significance was identified, pairwise comparisons using Dwass–Steel–Critchlow–Fligner (DSCF) tests [45] were performed to identify where differences lay (Table 3).

For the regression, we opted to utilize ordinal logistic regression (OLR); this approach was chosen due to the prevalence of Likert scale data that are categorized as a special case of ordinal data [46]. In contrast to conventional multiple regression, OLR treats the dependent variable as an ordered categorical variable, based on the principle of cumulative odds [47]. We reported the Nagelkerke’s R^2^ (R^2^N) [48] to summarize the proportion of variance in the dependent variable attributed to each independent variable; R^2^N is an adjusted version of the Cox and Snell R^2^ [49] which rescales the statistic to cover the full zero to one range. The background characteristics of participants were accounted for in each OLR.

## 3. Results

At the end of the recruitment period, 333 women with PCOS met the eligibility criteria and had fully completed the online questionnaires. The complete characteristics of the participants are reported in Table 1. In brief, the majority of participants self-reported that they were of White ethnic background (92.5%), had no children (73%), were in full-time employment (63.1%), and were in the lowest (≤£39,999) UK household income group (51.1%). In addition, 40% were married, and a further 40% were single, with the remainder being in other non-married relationships, divorced, separated or widowed. When participants self-reported their PCOS phenotype, 46.5% reported that they had been diagnosed with all three diagnostic characteristics (i.e., hyperandrogenism, menstrual disruption, and polycystic ovaries (PCO)), 13.5% with menstrual disruption and PCO, and 6.9% with menstrual disruption and hyperandrogenism, whilst the remaining participants were unsure about the phenotype related to their diagnosis [25].

When participants were split according to their categorical IPAQ PA levels (low PA: *n* = 100; moderate PA: *n* = 132; high PA: *n* = 101), there were no between group differences in age or years since their PCOS diagnosis (Table 2). Similarly, no statistical differences were found for any domain of the PCOSQOL between groups. However, significant differences were observed for BMI, with individuals performing high levels of PA having statistically lower BMI than the low and moderate PA groups (Table 3). When domains from the DASS-21 were analysed, it was only depression that presented any significant differences, with those in the high PA group having statistically lower levels of depression compared to the low and moderate PA groups (Table 3). Favourable differences for the high PA group were also observed for mental wellbeing (versus low and moderate PA), health today (versus low PA); illness perception (versus low PA), and insomnia severity (versus low PA); whilst illness perception was also lower in the moderate PA group compared to the low PA group.

When Kruskal–Wallis results comparing IPAQ classification groups were analysed (Table 2), there was statistical significance (*p* < 0.001) in each COM-B domain; DSCF pairwise comparisons (Table 3) further revealed that when those in the high PA group were compared to low (*p* ≤ 0.002) and moderate (*p* ≤ 0.049) PA groups, all comparisons statistically favoured those performing more PA. When the moderate was compared to the low PA group, statistical differences (*p* ≤ 0.022) were found for all domains apart from social and physical opportunity. In the OLR (Table 4), each of the COM-B domains, BMI and sitting time, were revealed to be significant predictors (*p* < 0.001) for PA categorisation. Conscious (OR: 1.44 (1.32, 1.57), R^2^N: 0.139, *p* < 0.001), and automatic (OR: 1.41 (1.29, 1.55), R^2^N: 0.120, *p* < 0.001) motivation were the two variables that explained the highest levels of variance.

## 4. Discussion

It has been widely reported in the general population that increasing PA participation reduces the risk of morbidity and mortality [50,51,52]. Although the potential benefits of participation in exercise are reportedly comparable in women with and without PCOS [20], women with PCOS have a less favourable health profile, despite many studies reporting no difference in the volume of exercise/PA performed compared to women without PCOS [53,54,55]. Whilst our previous systematic review [22] demonstrated that when compared to control, exercise interventions have favourable effects in women with PCOS, there is less clarity as to whether higher levels of PA have a beneficial effect in this population group. Accordingly, the first objective of the current study was to determine whether women with PCOS who self-reported performing higher levels of PA perceived their health and QoL more favourably than those reporting lower levels.

When PA categorical groups based upon participant IPAQ responses (i.e., low, moderate, or high) were created, between group statistical differences were identified for BMI, depression, health today, illness perception, mental wellbeing, and insomnia severity. Participants categorised in the highest PA group had favourable values (*p* ≤ 0.013) for all six of those outcomes when compared to those performing the lowest volume of PA. When the highest PA group were compared to the moderate PA group, there was still a statistically favourable effect (*p* ≤ 0.042) for BMI, depression, and mental wellbeing; when comparing the moderate to the low PA group, it was only illness perception that was lower when more PA was performed. Despite a lack of statistical significance for the remaining outcomes, the general trend was towards favourable results as PA levels increased.

As it would probably be expected, when BMI was compared between these groups, it was revealed that BMI was lower in the groups reporting more PA. Typically, a clinically important difference in body weight is defined as ≥5% [56], which, at least in the general population, is associated with improvements to cardiometabolic health [57,58]. Although the present study reports BMI as opposed to body weight, comparing the most active individuals to those performing moderate (−10.5%) or low (−18.3%) amounts of PA shows differences greater than 5%. Of note, the clinical relevance of these differences may be better quantified when BMI classifications are also taken into account [59]; thus, based on median scores, participants in the high PA group are classified in a lower category of obesity (class I obesity) than their counterparts in the moderate and low PA groups (class II obesity). Although a recent overview of systematic reviews [60] reported favourable changes in weight following exercise interventions, the magnitude of change (mean difference ranged from −1.5 to −3.5 kg; four systematic reviews incorporating 64 studies) is unlikely to reflect the between group differences reported in the current study. This means that there is some uncertainty as to whether the most active women with PCOS have a lower BMI because of the PA that they complete, or that they are more active because they have a lower BMI. Large prospective studies are still required to explore these research questions in women with PCOS. Interestingly, a previous study ascertained that obesity was a barrier to PA participation in a representative sample (*n* = 2298), and that this effect was strengthened in women with a BMI over 25 kg/m^2^ [61]. Moreover, overweightness and obesity are also associated with impaired QoL [62] with a greater impact in women [63], whilst reduced QoL, including psychological morbidity [64], poor health [61], and physical discomfort [65] have been cited as barriers to PA participation. As such, the findings of the current study tend to further support this notion, with those participants performing the highest levels of PA reporting higher levels of self-rated health, lower illness perception and lower depression levels than those performing lower levels of PA.

To gain a better understanding of the causal direction of these relationships, it is necessary to consider the secondary objective of the current study, that is whether self-reported capability, opportunity, and motivation can be used to predict PA participation in women with PCOS. Statistical differences for each COM-B domain were revealed when IPAQ categories were compared; generally, COM-B domain scores increased as the PA level/category increased in these three groups. In the OLR, all six domain scores demonstrated statistical significance when predicting IPAQ classification; reflective and automatic motivation were the strongest predictors, followed by physical and psychological capability, and then social and physical opportunity.

Motivation is a key determinant of sustained behaviour and is therefore a central focus to many behaviour change interventions [66]. Indeed, motivation, or lack thereof, is typically cited as a common barrier to PA participation, which appears to have more influence in females who have a BMI over 25 kg/m^2^ [61]. Of note, it has been suggested that motivational factors affecting women with PCOS are linked with risk perception [67], namely with the expectation of adverse events due to a given behaviour, and the severity of said event should it occur [68]. In this context, Ee and colleagues [67] have suggested that women with PCOS were concerned about injury and perceived exercise as tiring. This appears to be supported by our finding that women with the lowest PA levels had the lowest levels of automatic motivation. Lim and colleagues [69] reported that motivational barriers to exercise may also include tiredness and or feeling unrewarded by participation; our results revealed that sleep quality was statistically reduced in those performing the least PA, which may be influencing motivation to be more active. Compared to the general population, the women with PCOS in this study reported higher levels of sleep disruption during the COVID-19 pandemic [70], which may be a consequence of changes to routine [71] or concerns about the pandemic and its impact on their lives [71]. By contrast, there is a reported association between PA levels and sleep quality, with increasing PA having favourable effects on sleep quality in the general population [72,73], which may also be apparent in women with PCOS [74]. Consequently, the impact of fatigue/tiredness, and the extent of the potential bi-directional relationship between PA and sleep in PCOS is uncertain and warrants further investigation.

Regarding feelings of reward, it is unlikely that, compared to pre-COVID-19, beyond improvements to physical and mental health, these women would receive any additional tangible reward from increasing PA levels. However, with the disruption to healthcare services, a notable source of behavioural reinforcement and encouragement, coupled with an inability to meet face-to-face with friends and family, an important facilitator to PA participation, was restricted [69]. This reportedly caused significant concern amongst these women, with many resorting to unhealthy lifestyle practices as a direct result [75].

When considering the COVID-19 pandemic, perhaps the most striking recollections are of the disruption caused to daily routines, changes in mood, and the frustration that resulted from the lockdown restrictions [75,76]. As lockdown restrictions may have created more physical opportunity (i.e., time), and the UK’s Chief Medical Officer permitted an hour per day of outdoor exercise even at the peak of the pandemic (highlighting the importance of regular PA), it may be anticipated that PA might have increased [77]. However, the closure of gyms/leisure centres, alongside restrictions on all but essential travel, decreased the physical opportunity, which contributed to a reduction in PA for many individuals [78]. Indeed, women with PCOS reported that these closures, alongside changes to routine, had severely impacted their ability to manage their physical and mental wellbeing [75]. A major concern is that any changes to behaviours, and the associated health consequence, may persist beyond the pandemic; to better understand this phenomenon and tailor the appropriate support, there is a need to understand changes to the PA patterns (both in terms of volume and modality) of women with PCOS once all restrictions were lifted.

Similarly, social opportunity was also impacted during the lockdown. As previously stated, social mixing was prohibited during the lockdown; given the role of cohesion and social support in performing/maintaining PA behaviours [79], the inability to attend group exercise classes, or to engage with friends whilst performing PA, may also have served to reduce routine PA behaviours. However, beyond exercise intensity, it is unknown in the current study how and where participants were performing their PA, and the degree of importance placed upon these interactions. What has been documented is that some women with PCOS reported reduced anxiety because of no social contact; this was largely due to the perceived stigma and social pressures that are associated with increased body weight and not meeting societal beauty standards [75]. It is possible that due to increased time and reduced societal pressures, women with PCOS began exercising independently whilst at home, but addressing this question was beyond the scope of the present study.

Finally, physical and psychological capability were also able to statistically predict the PA classification within the present study, which is also in support of the point that women with PCOS who reported the lowest PA levels may be less active due to their higher BMI, degree of morbidity, and illness perception. Moreover, in addition to affecting motivation, poor sleep and the resulting fatigue may also contribute to reduced physical capability for PA in women with PCOS [69]. This is also in line with our findings that reported PA levels were higher with increasing sleep quality.

Interestingly, Ee and colleagues [67] also presented evidence that women with PCOS have varying levels of concern about their long-term metabolic health [80,81,82]. However, where perceived risk for weight gain and morbidity are increased, following a healthy lifestyle (including exercise/PA) was seen as less beneficial for weight gain prevention [83]. This may be, at least in part, a consequence of previously improving lifestyle habits, but failing to achieve significant weight loss [84]. Although the present study revealed that those performing the least PA also reported statistically higher illness perception, it is also possible that risk perception was increased due to the COVID-19 pandemic, the mandated stay at home orders, and participants’ concerns about the consequences of COVID-19 infection [85], since individuals with increased body weight and other comorbidities, including PCOS, are at increased risk of more severe COVID-19 [24,86]. Therefore, it is possible that women with PCOS and higher BMI who reported the lowest amount of PA (i.e., those at the greatest risk) were, to some extent, less active due to their risk perception about COVID-19 [75] However, the exact extent to which COVID-19 may have influenced risk perceptions during the pandemic lockdown in this population remains unclear and further studies are warranted to address this question.

Overall, by identifying motivation as the greatest predictor of PA levels, the present study indicates that there is a need to devise strategies to foster motivation towards increasing PA levels in women with PCOS. One such strategy can be the use of educational programmes to positively influence risk perceptions in these women by providing information on the consequences of being physically active/inactive [87], as well as examples of where others have achieved success [67]. Notably, it is important that the heterogeneity of PCOS is also accounted for, and that lifestyle interventions are also bespoke to the individual’s health priorities (e.g., cardiometabolic risk, fertility, mental health). With this in mind, it should be highlighted that women with PCOS should be involved in the decision-making processes for their care [88]. Indeed, such patients should undoubtedly be given the best available evidence on the effectiveness of treatment options but should then work with their healthcare practitioners to identify PA behaviours that they might find enjoyable rather than just those that are often incorrectly associated with exercise (e.g., going for a walk vs. running on a treadmill) [67].

### Study Limitations

There are certain limitations in the present study that are pertinent to acknowledge. Given the COVID-19 pandemic and the national restrictions that were in force during the study, data collection was conducted remotely and relied upon participant self-reporting. Whilst unavoidable in this instance, self-reported data may lead to decreased accuracy in the provided answers, which increases the risk of information bias in the study findings [89]. For example, when asked to self-report, individuals may tend to over-report their height and under-report their weight [90]; this effect appears to be further exaggerated in individuals with overweightness/obesity [91]. The implication in the current study is for potential discrepancies in the anthropometric characteristics of the study sample, which, given the key role of metabolic health in PCOS [92] and sleep quality [93] may influence the study findings. Moreover, regarding lifestyle behaviours, it has been reported that participants tend to provide responses that they believe are more socially desirable/acceptable, as opposed to a true reflection of their behaviours [94]. In the context of the present study, social desirability bias increases the risk that participants may have over-reported their PA levels [95] and under-reported their sedentary behaviours [96]. However, a methodology incorporating self-reported data is frequently employed in studies of this nature, whilst the relatively large study sample and the use of validated questionnaires to capture key study data may negate some of this potential bias. Future studies conducted at a time when there are no pandemic restrictions may wish to include a methodology that allows device-measurement of relevant outcomes.

Another potential study limitation is that comparative baseline data for the study cohort are lacking. However, the study survey questions asked about changes for a range of key outcomes due to COVID-19 restriction measures; without validated baseline measurements, it is not possible to quantify the exact extent to which the corresponding outcomes have been affected. Whilst it is not possible to retrospectively collect data from a time before the COVID-19 pandemic, this issue could somewhat be addressed by utilising a follow-up data collection now that the COVID-19 related restrictions have been lifted. To this aim, the relevant prospective follow-up of this study cohort has been planned.

Finally, although the relatively large sample size of the present study is a distinct strength, it is also accompanied by some inherent limitations. Indeed, although a common method in these type of studies, given the lockdown circumstances, the present study recruited a convenience sample, meaning that there are unknown factors about how representative the self-selected participants are of the population of interest, and that heterogeneity and/or outliers in the sample may affect the outcome of analyses. In the present study, responses were incomplete for 281 participants who started to complete the study online survey. This meant that these participants withdrew their consent to participate and thus were excluded from analyses. This was necessary to comply with the corresponding ethics, since based on the online nature of the study, non-completion of the survey was considered as withdrawal of consent to participate. Whilst this may introduce attrition bias, it was not possible to analyse whether there were systematic differences between those who completed the survey and those who did not [97]; given that participants were only tasked with completing a survey, this may minimise any potential bias in the results. It may however be indicative that there were issues (e.g., time to complete or participant comprehension) with the survey that was utilised.

Furthermore, the OLR analysis yielded statistically significant R^2^N for all variables, which may represent important findings but could also be a product of the large sample size fallacy [98]. The significant findings of the COM-B analysis in the present study certainly warrant further investigation. This could be conducted using an instrument that is even more sensitive to the COM-B constructs, and/or augmented with qualitative data to explore findings further. Finally, given the cross-sectional nature of the present study, the findings cannot be used to infer conclusions regarding the temporal/causal relationship between the outcomes. Thus, we have explored potential explanations for the observed effects, but we acknowledge that alternative study designs would be needed to confirm our findings.

## 5. Conclusions

The present study offers unique data that revealed that women with PCOS who self-reported performing higher levels of PA during the COVID-19 national lockdown in the UK tended to have favourable values for a range of factors, including lower BMI, depression, illness perception, and insomnia severity levels, as well as higher mental health and self-rated health. Moreover, the present findings indicate that, of the COM-B components, reflective and automatic motivation are the best predictors of PA level classification during the COVID-19 lockdown period in the UK. Although further large prospective studies are required to confirm and further explore these findings, it appears that future interventions for increasing PA in women with PCOS should crucially incorporate components targeting the individual’s motivation through education, risk mitigation, and intervention co-design.

## Figures and Tables

**Table 1 ijerph-20-02309-t001:** Selected sociodemographic characteristics for the study cohort of UK women with polycystic ovary syndrome.

Variable	*n* (%)
Ethnicity	
White	308 (92.5)
Mixed background	9 (2.7)
Asian or Asian British	8 (2.4)
Black or Black British	4 (1.2)
Other ethnic background	3 (0.9)
Declined to indicate	1 (0.3)
Relationship status	
Single	134 (40.2)
Married	132 (39.6)
Co-habiting	22 (6.6)
Long-term relationship	19 (5.7)
Civil partnership	12 (3.6)
Engaged	8 (2.4)
Divorced	3 (0.9)
Separated	2 (0.6)
Widowed	1 (0.3)
Children	
No	242 (72.7)
Yes	91 (27.3)
Education	
Undergraduate	135 (40.2)
College	107 (32.1)
Postgraduate	60 (18.0)
Secondary	26 (7.8)
Doctorate	5 (1.5)
Employment	
Full-time employment	210 (63.1)
Part-time employment	47 (14.1)
Student	27 (8.1)
House person	19 (5.7)
Unemployed	21 (6.3)
Self-employed	9 (2.7)
Household income	
≤£39,999	170 (51.1)
£40,000–£79,999	137 (41.1)
≥£80,000	26 (7.8)

All percentage data rounded to one decimal place.

**Table 2 ijerph-20-02309-t002:** Study outcome descriptive statistics (median (IQR)) for the whole study cohort (*n* = 333), and also categorised based upon their physical activity classification (low; moderate; high) according to the International Physical Activity Questionnaire (IPAQ).

Study Outcomes	Full Study Cohort of Women with PCOS (*n* = 333)	Physical Activity Classification			Kruskal-Wallis		
		Low (*n* = 100)	Moderate (*n* = 132)	High (*n* = 101)	χ^2^	ε^2^	*p*
Age (years)	30.00 (9.00)	29.00 (12.00)	30.00 (9.00)	30.00 (8.00)	1.20	0.004	0.548
Years Since PCOS Diagnosis	8.00 (9.70)	6.90 (9.75)	7.40 (9.50)	8.40 (10.70)	3.38	0.010	0.185
BMI (kg/m^2^)	34.80 (13.60)	38.70 (13.40)	35.30 (11.20)	31.60 (10.30)	27.43	0.084	<0.001
PCOSQOL							
Impact of PCOS	40.00 (28.00)	35.00 (28.50)	40.00 (29.00)	43.00 (29.30)	5.12	0.015	0.077
Infertility	24.00 (27.00)	21.00 (24.80)	26.00 (27.00)	25.50 (27.30)	1.32	0.004	0.516
Hirsutism	16.00 (18.00)	14.50 (16.00)	15.00 (14.00)	17.00 (20.50)	4.94	0.015	0.084
Mood	17.00 (10.00)	17.00 (12.30)	18.00 (9.00)	17.00 (8.00)	0.60	0.002	0.743
Total score	101.00 (60.00)	90.50 (59.30)	101.00 (60.00)	108.00 (59.30)	4.47	0.013	0.107
DASS-21							
Depression	9.00 (8.00)	11.00 (9.00)	9.00 (8.00)	7.00 (6.00)	15.44	0.047	<0.001
Anxiety	5.00 (6.00)	6.00 (8.00)	5.00 (6.00)	5.00 (5.00)	3.61	0.011	0.164
Stress	9.00 (7.00)	10.50 (7.50)	9.00 (6.00)	9.00 (6.30)	1.75	0.005	0.416
Health Today	66.00 (26.00)	60.00 (32.50)	65.00 (25.00)	70.00 (25.00)	16.08	0.048	<0.001
Brief Illness Perception	54.00 (13.00)	57.50 (15.00)	54.00 (12.00)	52.00 (11.00)	16.30	0.049	<0.001
Mental Wellbeing	38.00 (10.00)	35.00 (11.00)	37.00 (10.00)	40.00 (10.00)	19.25	0.058	<0.001
Insomnia Severity Index	12.00 (9.00)	13.00 (9.00)	12.00 (10.00)	11.00 (9.00)	8.00	0.024	0.018
COM-B							
Psychological capability	6.00 (4.00)	4.00 (4.00)	6.00 (4.00)	7.00 (4.00)	31.07	0.094	<0.001
Physical capability	7.00 (3.25)	6.00 (3.00)	7.00 (3.00)	8.00 (3.00)	28.50	0.086	<0.001
Social opportunity	6.00 (3.00)	4.50 (4.00)	6.00 (3.00)	6.50 (3.00)	21.16	0.064	<0.001
Physical opportunity	6.00 (3.00)	5.00 (5.00)	6.00 (3.00)	7.00 (3.00)	12.88	0.039	<0.001
Automatic motivation	4.00 (4.00)	2.00 (2.25)	4.00 (3.00)	5.00 (4.00)	55.64	0.169	<0.001
Reflective motivation	4.00 (4.00)	2.00 (3.00)	4.00 (4.00)	6.00 (4.00)	65.46	0.198	<0.001
IPAQ							
Total MET-mins/week	1851.00 (2922.00)	396.00 (665.00)	1520.00 (900.00)	4404.00 (2463.00)	243.90	0.735	<0.001
Vigorous MET-mins/wk	480.00 (1440.00)	0.00 (160.00)	0.00 (640.00)	1920.00 (1920.00)	153.80	0.463	<0.001
Moderate MET-mins/wk	80.00 (720.00)	0.00 (0.00)	240.00 (480.00)	670.00 (1440.00)	78.80	0.237	<0.001
Walking MET-mins/wk	792.00 (1251.00)	198.00 (446.00)	924.00 (891.00)	1386.00 (2030.00)	126.20	0.380	<0.001
Sitting hrs/day	8.00 (5.00)	10.0 (5.63)	8.00 (6.00)	6.00 (4.50)	40.40	0.122	<0.001

Key: PCOS: polycystic ovary syndrome; N: number of participants; χ^2^: chi-square test of expected frequencies; ε^2^: partial eta-square; *p*: statistical significance; BMI: body mass index; PCOSQOL: PCOS Quality of Life-Questionnaire; DASS-21: Depression, Anxiety, and Stress Scale (21-item); COM-B: Capability, Opportunity, Motivation, Behaviour Questionnaire; IPAQ: International Physical Activity Questionnaire—Short Form; MET: metabolic equivalents of task. Kruskal–Wallis analyses were used to test for differences between IPAQ classification.

**Table 3 ijerph-20-02309-t003:** Dwass–Steel–Critchlow–Fligner (DSCF) pairwise comparisons for variables where statistical significance was found in the Kruskal–Wallis analyses (Table 1). Comparator groups represent categorical group membership based upon physical activity classification (low; moderate; high) according to the International Physical Activity Questionnaire (IPAQ).

Outcomes	Comparator One	Comparator Two	W	*p*
BMI (kg/m^2^)	Low	High	−7.16	<0.001
	Low	Moderate	−2.68	0.141
	Moderate	High	−5.43	<0.001
Depression (DASS-21)	Low	High	−5.41	<0.001
	Low	Moderate	−1.94	0.356
	Moderate	High	−3.42	0.042
Health Today	Low	High	5.64	<0.001
	Low	Moderate	3.04	0.081
	Moderate	High	2.49	0.182
Brief Illness Perception	Low	High	−5.65	<0.001
	Low	Moderate	−3.72	0.023
	Moderate	High	−1.67	0.463
Mental Wellbeing	Low	High	5.82	<0.001
	Low	Moderate	1.91	0.367
	Moderate	High	4.25	0.007
Insomnia Severity Index	Low	High	−4.01	0.013
	Low	Moderate	−2.22	0.258
	Moderate	High	−1.61	0.491
COM-B				
Psychological capability	Low	High	7.61	<0.001
	Low	Moderate	3.76	0.022
	Moderate	High	4.38	0.006
Physical capability	Low	High	7.15	<0.001
	Low	Moderate	4.50	0.004
	Moderate	High	3.67	0.026
Social opportunity	Low	High	6.29	<0.001
	Low	Moderate	3.08	0.075
	Moderate	High	3.58	0.031
Physical opportunity	Low	High	4.73	0.002
	Low	Moderate	2.06	0.312
	Moderate	High	3.33	0.049
Automatic motivation	Low	High	10.23	<0.001
	Low	Moderate	5.32	<0.001
	Moderate	High	5.52	<0.001
Reflective motivation	Low	High	10.86	<0.001
	Low	Moderate	6.66	<0.001
	Moderate	High	5.69	<0.001

Key: BMI: Body mass index; DASS-21: Depression, Anxiety, and Stress Scale (21-item); COM-B: Capability, Opportunity, Motivation, Behaviour Questionnaire; W: Wilcoxon rank sum test statistic; *p*: statistical significance.

**Table 4 ijerph-20-02309-t004:** Ordinal logistic regression and predictive ability for physical activity classification (low; moderate; high) based on the International Physical Activity Questionnaire (IPAQ).

		95% CI					95% CI			
Predictor	Estimate	Lower	Upper	SE	Z	OR	Lower	Upper	R^2^N	*p*
Reflective motivation	0.36	0.28	0.45	0.05	8.01	1.44	1.32	1.57	0.139	<0.001
Automatic motivation	0.35	0.26	0.44	0.05	7.48	1.41	1.29	1.55	0.120	<0.001
Physical capability	0.24	0.15	0.33	0.05	5.28	1.27	1.16	1.39	0.056	<0.001
Psychological capability	0.23	0.15	0.31	0.04	5.69	1.26	1.16	1.36	0.066	<0.001
Social opportunity	0.18	0.10	0.26	0.04	4.46	1.20	1.11	1.30	0.040	<0.001
Physical opportunity	0.15	0.07	0.24	0.04	3.65	1.17	1.07	1.27	0.026	<0.001
BMI (kg/m^2^)	−0.06	−0.09	−0.04	0.01	−5.31	0.94	0.92	0.96	0.058	<0.001
Sitting time (hrs/day)	−0.17	−0.22	−0.11	0.03	−5.99	0.85	0.80	0.89	0.077	<0.001

## Data Availability

The raw data supporting the conclusions of this article will be made available, without undue reservation, by the authors upon reasonable request and where it is ethically acceptable to do so, and does not violate the protection of participants, or other valid ethical, privacy or security concerns.

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
