# Peer review of "Capability, Opportunity, and Motivation—Identifying Constructs for Increasing Physical Activity Behaviours in Women with Polycystic Ovary Syndrome (PCOS)"

_ijerph, 2023, doi:10.3390/ijerph20032309_

Round 1

Reviewer 1 Report

In my opinion, this is a very well-written manuscript which investigates whether women who self-reported being more physically active perceived their health and QoL to be better than those who were less active; and, second, whether constructs of the COM-B model can be used to predict variance in PA behaviour so that these individual constructs could be targeted in future behavioural interventions to increase PA in women with PCOS. However, I just have two comments:

1.     Please update the abstract according to the guideline and clearly state this study's objective.

2.     How do you get/calculate the number of participants for this study?

Author Response

Reply:  We thank the reviewer for the time dedicated to reviewing our paper and for the positive feedback and helpful comments/suggestions. Accordingly, we have made the following revisions based on the provided comments:

1) As suggested, we have updated the abstract according to the journal’s guidelines (please see the updated abstract). The abstract headings have been removed and the total wordcount has been reduced to 200 words. We have also included the study aim/purpose (please see lines 28-30).

2)  We have now added into the manuscript that this was a convenience sample which was recruited via social media during the lockdown (please see lines 111-115). We have also acknowledged this as a limitation of the study design (please see lines 449-453).

Reviewer 2 Report

The article deals with a very important topic which is polycystic ovary syndrome. Unfortunately, the article lacks the exact characteristics of the study group, comparing the results obtained from the implementation of the research. The conclusions are very general. I believe that such oral issues raised in the article should be described in more detail. The results obtained Well-designed results can be used in the dissemination of information and PCOS prevention.

I believe that the article should be re-edited, supplemented with important details that the authors are sure to have, and resubmitted for publication.

Author Response

Reply:  We thank the reviewer for the time dedicated to reviewing our paper and for the helpful comments/suggestions. We have made the following revisions according to the provided comments:

1) We have added in a new table (please see Table 1), which reports the sociodemographic information for the whole study sample.

2) The discussion has been substantially updated so as to better cover the findings surrounding the COM-B model and physical activity. This now also includes information which is specific to the COVID-19 pandemic and this at-risk population (please see lines 335-381).

Reviewer 3 Report

Reviewer's comments:

Abstract:

The abstract should reduce the number of words to 200, according to instructions for authors from IJERPH. In the submitted format, it contains 274 words.

Introduction:

On lines 80 - 84, I suggest explaining further how COVID-19 can be an additional risk for women with PCOS. The referenced articles that support this statement only report that due to the cardiometabolic conditions of women with PCOS, COVID-19 may make them more vulnerable, however they do not describe the pathophysiology related to additional risks.

I suggest authors revise and modify the dual purpose of lines 100-105. The aim of studies is always important to be general and comprehensive. I believe that the authors confused the study hypothesis with objectives. Based on the hypothesis, it is understood that first they would like to confirm whether women who declared themselves to be more physically active perceived their health and quality of life as better than those who were less active, if this hypothesis is confirmed, they would investigate constructions of the COM-B model. in AF behavior. The objective understood by the study is to identify constructs to increase physical activity behaviors in women with polycystic ovary syndrome (PCOS).

Materials and Methods: 

Are described in sufficient detail to allow others to replicate and build on the published results. However, what are the exclusion criteria applied in the study? It is important that the method contains the total number of the sample plus the number of excluded participants and the reasons for exclusion.

Results: 

Concise and precise description of experimental results with interpretation and conclusions prior to discussion.

Discussion:

I suggest further discussion about the second objective of the study, starting with line 294, as the arguments are shallow and superficial.

Conclusion:

The authors can be more concise, actually bringing in the conclusion to the answer to the proposed objectives, about what the COM-B components can help in the behavior and interventions of women with PCOS.

Author Response

Reply:  We thank the reviewer for the time dedicated to reviewing our paper and for the positive feedback and helpful comments/suggestions. Accordingly, we have made the following revisions based on the provided comments for each section:

1) Abstract: As suggested, we have updated the abstract according to the journal’s guidelines and stated the study's objective (please see the updated abstract).

2)   Introduction: We have presented some specific risk factors for women with PCOS which may predispose them to severe COVID-19. Furthermore, we have cited additional papers which also explore the pathophysiology behind this risk (please see lines 82-87, and added references 26-28).

3) Aims and objectives: We have reworded the aims. There is now a clear hypothesis which is followed by additional aims around COM-B and PA variance (please see lines 103-108).

4) Methods: As requested, we have added the exclusion criteria to the methods (please see lines 116-118). Furthermore, we mention the total number of participants that confirmed that they were eligible to participate in the study (n=614) and that 281 of these were excluded from the study analyses because they did not complete all questions (please see lines 126-128). Based on the online nature of the present study and to comply with the corresponding ethics approval, non-completion of the online survey was considered as withdrawal of consent to participate. This is also now mentioned in the limitations section (please see lines 454-459).

5) Discussion: We have added more information to the discussion. According to the reviewer’s recommendations, this focuses on the COM-B domains and explores factors relating to COVID-19 lockdown and also lifestyle behaviors in women with PCOS (please see lines 335-381).

6) Conclusion: As suggested, we have made the conclusion more concise (we have removed some points which were already mentioned in the limitations section), stating that based on the present findings (which should be confirmed and further explored by future larger studies) future interventions for increasing PA in women with PCOS should target the motivation of women with PCOS to participate in PA (please see lines 476-486 for the updated and abbreviated conclusion section).

Reviewer 4 Report

The manuscript titled with ‘Capability, Opportunity, and Motivation - identifying constructs for increasing physical activity behaviours in women with polycystic ovary syndrome (PCOS)’, written by Chris Kite et al, is over-all well written, clearly and scientifically demonstrated, properly concluded with objectively listed self-sensed limitations.

However, from my perspective, there seem to be a logical problem needs to be improved or explained.

From the title and the described aim of this study, identifying the COM-B components for increasing PA in PCOS women are the most important parts and the objective of the study. However, in the section of Materials and Methods, the COM-B questionnaire was not introduced, in the interpretation of Results as well as Discussion, COM-B was not adequately mentioned/discussed, instead, other factors like metal, emotional, perceptional, BMI, insomnia factors were largely described and discussed. What I expected upon reading the Title and Introduction is a further detailed investigation on COM-B. Even a self-designed detailed questionnaire fitting the study aim was expected (e.g., which kinds of physical or psychological or social factors are mostly encountered in COM-B, and discuss about what could be done to improve this situation, et al.). Since effects of PA on PCOS patients or even healthy people are not a novel topic, but to go deeper into what determines/promotes/hinders the ‘administration’ of PA in POCS patients is very interesting and meaningful to investigate. 

Author Response

Reply:  We thank the reviewer for the time dedicated to reviewing our paper and for the positive feedback and helpful comments/suggestions. Accordingly, we have made the following revisions based on the provided comments:

1) COM-B and the questionnaire: As requested, the COM-B framework is now clearly mentioned in the introductory section (please see lines 92-103) and links drawn to the importance of being physically active are made.

2) Discussion: We have added more information to the discussion. According to the overall recommendations that we received, this focuses on the COM-B domains and explores factors relating to COVID-19 lockdown and also to lifestyle behaviors in women with PCOS (please see lines 335-381).